# Quantized Kernel Learning for Feature Matching

**Danfeng Qin**
ETH Zürich

**Xuanli Chen**
TU Munich

**Matthieu Guillaumin**
ETH Zürich

**Luc Van Gool**
ETH Zürich

{qind, guillaumin, vangool}@vision.ee.ethz.ch, xuanli.chen@tum.de

## Abstract

Matching local visual features is a crucial problem in computer vision and its accuracy greatly depends on the choice of similarity measure. As it is generally very difficult to design by hand a similarity or a kernel perfectly adapted to the data of interest, learning it automatically with as few assumptions as possible is preferable. However, available techniques for kernel learning suffer from several limitations, such as restrictive parametrization or scalability.

In this paper, we introduce a simple and flexible family of non-linear kernels which we refer to as Quantized Kernels (QK). QKs are arbitrary kernels in the index space of a data quantizer, i.e., piecewise constant similarities in the original feature space. Quantization allows to compress features and keep the learning tractable. As a result, we obtain state-of-the-art matching performance on a standard benchmark dataset with just a few bits to represent each feature dimension. QKs also have explicit non-linear, low-dimensional feature mappings that grant access to Euclidean geometry for uncompressed features.

## 1 Introduction

Matching local visual features is a core problem in computer vision with a vast range of applications such as image registration [28], image alignment and stitching [6] and structure-from-motion [1]. To cope with the geometric transformations and photometric distorsions that images exhibit, many robust feature descriptors have been proposed. In particular, histograms of oriented gradients such as SIFT [15] have proved successful in many of the above tasks. Despite these results, they are inherently limited by their design choices. Hence, we have witnessed an increasing amount of work focusing on automatically learning visual descriptors from data via discriminative embeddings [11, 4] or hyper-parameter optimization [5, 21, 23, 22].

A dual aspect of visual description is the measure of visual (dis-)similarity, which is responsible for deciding whether a pair of features matches or not. In image registration, retrieval and 3D reconstruction, for instance, nearest neighbor search builds on such measures to establish point correspondences. Thus, the choice of similarity or kernel impacts the performance of a system as much as the choice of visual features [2, 16, 18]. Designing a good similarity measure for matching is difficult and commonly used kernels such as the linear, intersection, $\chi^2$ and RBF kernels are not ideal as their inherent properties (*e.g.*, stationarity, homogeneity) may not fit the data well.

Existing techniques for automatically learning similarity measures suffer from different limitations. Metric learning approaches [25] learn to project the data to a lower-dimensional and more discriminative space where the Euclidean geometry can be used. However, these methods are inherently linear. Multiple Kernel Learning (MKL) [3] is able to combine multiple base kernels in an optimal way, but its complexity limits the amount of data that can be used and forces the user to pre-select or design a small number of kernels that are likely to perform well. Additionally, the resulting kernel may not be easily represented in a reasonably small Euclidean space. This is problematic, as many efficient algorithms (*e.g.* approximate nearest neighbor techniques) heavily rely on Euclidean geometry and have non-intuitive behavior in higher dimensions.

In this paper, we introduce a simple yet powerful family of kernels, *Quantized Kernels* (QK), which (a) model non-linearities and heterogeneities in the data, (b) lead to compact representations that can be easily decompressed into a reasonably-sized Euclidean space and (c) are efficient to learn so that large-scale data can be exploited. In essence, we build on the fact that vector quantizers project data into a finite set of $N$ elements, the *index space*, and on the simple observation that kernels on finite sets are fully specified by the $N \times N$ Gram matrix of these elements (the *kernel matrix*), which we propose to learn directly. Thus, QKs are piecewise constant but otherwise arbitrary, making them very flexible. Since the learnt kernel matrices are positive semi-definite, we directly obtain the corresponding explicit feature mappings and exploit their potential low-rankness.

In the remainder of the paper, we first further discuss related work (Sec. 2), then present QKs in detail (Sec. 3). As important contributions, we show how to efficiently learn the quantizer and the kernel matrix so as to maximize the matching performance (Sec. 3.2), using an exact linear-time inference subroutine (Sec. 3.3), and devise practical techniques for users to incorporate knowledge about the structure of the data (Sec. 3.4) and reduce the number of parameters of the system. Our experiments in Sec. 4 show that our kernels yield state-of-the-art performance on a standard feature matching benchmark and improve over kernels used in the literature for several descriptors, including one based on metric learning. Our compressed features are very compact, using only 1 to 4 bits per dimension of the original features. For instance, on SIFT descriptors, our QK yields about 10% improvement on matching compared to the dot product, while compressing features by a factor 8.

## 2   Related work

Our work relates to a vast literature on kernel selection and tuning, descriptor, similarity, distance and kernel learning. We present a selection of such works below.

**Basic kernels and kernel tuning.**   A common approach for choosing a kernel is to pick one from the literature: dot product, Gaussian RBF, intersection [16], $\chi^2$, Hellinger, etc. These generic kernels have been extensively studied [24] and have properties such as homogeneity or stationarity. These properties may be inadequate for the data of interest and thus the kernels will not yield optimal performance. Efficient yet approximate versions of such kernels [9, 20, 24] are similarly inadequate.

**Descriptor learning.**   Early work on descriptor learning improved SIFT by exploring its parameter space [26]. Later, automatic parameter selection was proposed with a non-convex objective [5]. Recently, significant improvements in local description for matching have been obtained by optimizing feature encoding [4] and descriptor pooling [21, 23]. These works maximize the matching performance directly via convex optimization [21] or boosting [23]. As we show in our experiments, our approach improves matching even for such optimized descriptors.

**Distance, similarity and kernel learning.**   Mahalanobis metrics (*e.g.*, [25]) are probably the most widely used family of (dis-)similarities in supervised settings. They extend the Euclidean metric by accounting for correlations between input dimensions and are equivalent to projecting data to a new, potentially smaller, Euclidean space. Learning the projection improves discrimination and compresses feature vectors, but the projection is inherently linear.[1] There are several attempts to learn more powerful non-linear kernels from data. Multiple Kernel Learning (MKL) [3] operates on a parametric family of kernels: it learns a convex combination of a few base kernels so as to maximize classification accuracy. Recent advances now allow to combine thousands of kernels in MKL [17] or exploit specialized families of kernels to derive faster algorithms [19]. In that work, the authors combine binary base kernels based on randomized indicator functions but restricted them to XNOR-like kernels. Our QK framework can also be seen as an efficient and robust MKL on a specific family of binary base kernels. However, our binary base kernels originate from more general quantizations: they correspond to their regions of constantness. As a consequence, the resulting optimization problem is also more involves and thus calls for approximate solutions.

In parallel to MKL approaches, Non-Parametric Kernel Learning (NPKL) [10] has emerged as a flexible kernel learning alternative. Without any assumption on the form of the kernel, these methods aim at learning the Gram matrix of the data directly. The optimization problem is a semi-definite program whose size is quadratic in the number of samples. Scalability is therefore an issue, and approximation techniques must be used to compute the kernel on unobserved data. Like NPKL, we learn the values of the kernel matrix directly. However, we do it in the index space instead of the

original space. Hence, we restrict our family of kernels to piecewise constant ones[2], but, contrary to NPKL, the complexity of the problems we solve does not grow with the number of data points but with the refinement of the quantization and our kernels trivially generalize to unobserved inputs.

## 3 Quantized kernels

In this section, we present the framework of quantized kernels (QK). We start in Sec. 3.1 by defining QKs and looking at some of their properties. We then present in Sec. 3.2 a general alternating learning algorithm. A key step is to optimize the quantizer itself. We present in Sec. 3.3 our scheme for quantization optimization for a single dimensional feature and how to generalize it to higher dimensions in Sec. 3.4.

### 3.1 Definition and properties

Formally, *quantized kernels* $\mathcal{QK}_N^D$ are the set of kernels $k_q$ on $\mathbb{R}^D \times \mathbb{R}^D$ such that:

$$\exists q : \mathbb{R}^D \mapsto \{1, \dots, N\}, \quad \exists \mathbf{K} \in \mathbb{R}^{N \times N} \succeq 0, \quad \forall \mathbf{x}, \mathbf{y} \in \mathbb{R}^D, \quad k_q(\mathbf{x}, \mathbf{y}) = \mathbf{K}(q(\mathbf{x}), q(\mathbf{y})), \quad (1)$$

where $q$ is a quantization function which projects $\mathbf{x} \in \mathbb{R}^D$ to the finite index space $\{1, \dots, N\}$, and $\mathbf{K} \succeq 0$ denotes that $\mathbf{K}$ is a positive semi-definite (PSD) matrix. As discussed above, quantized kernels are an efficient parametrization of piecewise constant functions, where $q$ defines the regions of constantness. Moreover, the $N \times N$ matrix $\mathbf{K}$ is unique for a given choice of $k_q$, as it simply accounts for the $N(N+1)/2$ possible values of the kernel and is the Gram matrix of the $N$ elements of the index space. We can also see $q$ as a 1-of-$N$ coding feature map $\varphi_q$, such that:

$$k_q(\mathbf{x}, \mathbf{y}) = \mathbf{K}(q(\mathbf{x}), q(\mathbf{y})) = \varphi_q(\mathbf{x})^\top \mathbf{K} \varphi_q(\mathbf{y}). \quad (2)$$

The components of the matrix $\mathbf{K}$ fully parametrize the family of quantized kernels based on $q$, and it is a PSD matrix if and only if $k_q$ is a PSD kernel. An explicit feature mapping of $k_q$ is easily computed from the Cholesky decomposition of the PSD matrix $\mathbf{K} = \mathbf{P}^\top \mathbf{P}$:

$$k_q(\mathbf{x}, \mathbf{y}) = \varphi_q(\mathbf{x})^\top \mathbf{K} \varphi_q(\mathbf{y}) = \left\langle \psi_q^{\mathbf{P}}(\mathbf{x}), \psi_q^{\mathbf{P}}(\mathbf{y}) \right\rangle, \quad (3)$$

where $\psi_q^{\mathbf{P}}(\mathbf{x}) = \mathbf{P}\varphi_q(\mathbf{x})$. It is of particular interest to limit the rank $N' \leq N$ of $\mathbf{K}$, and hence the number of rows in $\mathbf{P}$. In their *compressed form*, vectors require only $\log_2(N)$ bits of memory for storing $q(\mathbf{x})$ and they can be decompressed in $\mathbb{R}^{N'}$ using $\mathbf{P}\varphi_q(\mathbf{x})$. Not only is this decompressed vector smaller than one based on $\varphi_q$, but it is also associated with the Euclidean geometry rather than the kernel one. This allows the exploitation of the large literature of efficient methods specialized to Euclidean spaces.

### 3.2 Learning quantized kernels

In this section, we describe a general alternating algorithm to learn a quantized kernel $k_q$ for feature matching. This problem can be formulated as quadruple-wise constraints of the following form:

$$k_q(\mathbf{x}, \mathbf{y}) > k_q(\mathbf{u}, \mathbf{v}), \qquad \forall (\mathbf{x}, \mathbf{y}) \in \mathcal{P}, \quad \forall (\mathbf{u}, \mathbf{v}) \in \mathcal{N}, \quad (4)$$

where $\mathcal{P}$ denotes the set of *positive* feature pairs, and $\mathcal{N}$ is the *negative* one. The positive set contains feature pairs that should be visually matched, while the negative pairs are mismatches.

We adopt a large-margin formulation of the above constraints using the trace-norm regularization $\|\cdot\|_*$ on $\mathbf{K}$, which is the tightest convex surrogate to low-rank regularization [8]. Using $M$ training pairs $\{(\mathbf{x}_j, \mathbf{y}_j)\}_{j=1\dots M}$, we obtain the following optimization problem:

$$\underset{\mathbf{K} \succeq 0, \, q \in \mathcal{Q}_N^D}{\operatorname{argmin}} \quad E(\mathbf{K}, q) = \frac{\lambda}{2}\|\mathbf{K}\|_* + \sum_{j=1}^{M} \max\left(0, 1 - l_j \varphi_q(\mathbf{x}_j)^\top \mathbf{K} \varphi_q(\mathbf{y}_j)\right), \quad (5)$$

where $\mathcal{Q}_N^D$ denotes the set of quantizers $q : \mathbb{R}^D \mapsto \{1, \dots, N\}$, the pair label $l_j \in \{-1, 1\}$ denotes whether the feature pair $(\mathbf{x}_j, \mathbf{y}_j)$ is in $\mathcal{N}$ or $\mathcal{P}$ respectively. The parameter $\lambda$ controls the trade-off between the regularization and the empirical loss. Solving Eq. (5) directly is intractable. We thus propose to alternate between the optimization of $\mathbf{K}$ and $q$. We describe the former below, and the latter in the next section.

**Optimizing K with fixed** $q$. When fixing $q$ in Eq. (5), the objective function becomes convex in $\mathbf{K}$ but is not differentiable, so we resort to stochastic sub-gradient descent for optimization. Similar to [21], we used *Regularised Dual Averaging* (RDA) [27] to optimize $\mathbf{K}$ iteratively. At iteration $t+1$, the kernel matrix $\mathbf{K}_{t+1}$ is updated with the following rule:

$$\mathbf{K}_{t+1} = \Pi\left(-\frac{\sqrt{t}}{\gamma}\left(\overline{\mathbf{G}_t} + \lambda\mathbf{I}\right)\right) \tag{6}$$

where $\gamma > 0$ and $\overline{\mathbf{G}_t} = \frac{1}{t}\sum_{t'=1}^{t} \mathbf{G}_{t'}$ is the rolling average of subgradients $\mathbf{G}_{t'}$ of the loss computed at step $t'$ from one sample pair. $\mathbf{I}$ is the identity matrix and $\Pi$ is the projection onto the PSD cone.

### 3.3 Interval quantization optimization for a single dimension

To optimize an objective like Eq. (5) when $\mathbf{K}$ is fixed, we must consider how to design and parametrize the elements of $\mathcal{Q}_N^D$. In this work, we adopt *interval quantizers*, and in this section we assume $D = 1$, *i.e.*, restrict the study of quantization to $\mathbb{R}$.

**Interval quantizers.** An interval quantizer $q$ over $\mathbb{R}$ is defined by a set of $N+1$ boundaries $b_i \in \mathbb{R}$ with $b_0 = -\infty$, $b_N = \infty$ and $q(x) = i$ if and only if $b_{i-1} < x \leq b_i$. Importantly, interval quantizers are monotonous, $x \leq y \Rightarrow q(x) \leq q(y)$, and boundaries $b_i$ can be set to any value between $\max_{q(x)=i} x$ (included) and $\min_{q(x)=i+1} x$ (excluded). Therefore, Eq. (5) can be viewed as a data labelling problem, where each value $x_j$ or $y_j$ takes a label in $[1, N]$, with a monotonicity constraint.

Thus, let us now consider the graph $(\mathcal{V}, \mathcal{E})$ where nodes $\mathcal{V} = \{v_t\}_{t=1\ldots 2M}$ represent the list of all $x_j$ and $y_j$ in a sorted order and the edges $\mathcal{E} = \{(v_s, v_t)\}$ connect all pairs $(x_j, y_j)$. Then Eq. (5) with fixed $\mathbf{K}$ is equivalent to the following discrete pairwise energy minimization problem:

$$\operatorname*{argmin}_{\mathbf{q}\in[1,N]^{2M}} \quad E'(\mathbf{q}) = \sum_{(s,t)\in\mathcal{E}} E_{st}(q(v_s), q(v_t)) + \sum_{t=2}^{2M} C_t(q(v_{t-1}), q(v_t)), \tag{7}$$

where $E_{st}(q(v_s), q(v_t)) = E_j(q(x_j), q(y_j)) = \max(0, 1 - l_j\mathbf{K}(q(x_j), q(y_j)))$ and $C_t$ is $\infty$ for $q(v_t) < q(v_{t-1})$ and $0$ otherwise (*i.e.*, it encodes the monotonicity of $q$ in the sorted list of $v_t$).

The optimization of Eq. (7) is an NP-hard problem as the energies $E_{st}$ are arbitrary and the graph does not have a bounded treewidth, in general. Hence, we iterate the individual optimization of each of the boundaries using an exact linear-time algorithm, which we present below.

**Exact linear-time optimization of a binary interval quantizer.** We now consider solving equations of the form of Eq. (7) for the binary label case ($N = 2$). The main observation is that the monotonicity constraint means that labels are 1 until a certain node $t$ and then 2 from node $t+1$, and this switch can occur only once on the entire sequence, where $v_t \leq b_1 < v_{t+1}$. This means that there are only $2M+1$ possible labellings and we can order them from $(1, \ldots, 1)$, $(1, \ldots, 1, 2)$ to $(2, \ldots, 2)$. A naïve algorithm consists in computing the $2M+1$ energies explicitly. Since each energy computation is linear in the number of edges, this results in a quadratic complexity overall.

A linear-time algorithm exist. It stems from the observation that the energies of two consecutive labellings (*e.g.*, switching the label of $v_t$ from 1 to 2) differ only by a constant number of terms:

$$E(q(v_{t-1})=1, q(v_t)=2, q(v_{t+1})=2) = E(q(v_{t-1})=1, q(v_t)=1, q(v_{t+1})=2)$$
$$+ C_t(1,2) - C_t(1,1) + C_{t+1}(2,2) - C_{t+1}(1,2) + E_{st}(q(v_s),2) - E_{st}(q(v_s),1) \tag{8}$$

where, w.l.o.g., we have assumed $(s,t) \in \mathcal{E}$. After finding the optimal labelling, *i.e.* finding the label change $(v_t, v_{t+1})$, we set $b_1 = (v_t + v_{t+1})/2$ to obtain the best possible generalization.

**Finite spaces.** When the input feature space has a finite number of different values (*e.g.*, $x \in [1, T]$), then we can use linear-time sorting and merge all nodes with equal value in Eq. (7): this results in considering at most $T+1$ labellings, which is potentially much smaller than $2M+1$.

**Extension to the multilabel case.** Optimizing a single boundary $b_i$ of a multilabel interval quantization is essentially the same binary problem as above, where we limit the optimization to the values currently assigned to $i$ and $i+1$ and keep the other assignments $\overline{q}$ fixed. We use unaries $E_j(q(x_j), \overline{q}(y_j))$ or $E_j(\overline{q}(x_j), q(y_j))$ to model half-fixed pairs for $x_j$ or $y_j$, respectively.

### 3.4 Learning higher dimensional quantized kernels

We now want to generalize interval quantizers to higher dimensions. This is readily feasible via product quantization [13], using interval quantizers for each individual dimension.

**Interval product quantization.** An interval product quantizer $q(\mathbf{x}) : \mathbb{R}^D \mapsto \{1, \ldots, N\}$ is of the form $q(\mathbf{x}) = (q_1(\mathbf{x}_1), \ldots, q_D(\mathbf{x}_D))$, where $q_1, \ldots, q_D$ are interval quantizers with $N_1, \ldots, N_D$ bins respectively, i.e., $N = \prod_{d=1}^{D} N_d$. The learning algorithm devised above trivially generalizes to interval product quantization by fixing all but one boundary of a single component quantizer $q_d$. However, learning $\mathbf{K} \in \mathbb{R}^N \times \mathbb{R}^N$ when $N$ is very large becomes problematic: not only does RDA scale unfavourably, but the lack of training data will eventually lead to severe overfitting. To address these issues, we devise below variants of QKs that have practical advantages for robust learning.

**Additive quantized kernels (AQK).** We can drastically reduce the number of parameters by restricting product quantized kernels to additive ones, which consists in decomposing over dimensions:

$$k_q(\mathbf{x}, \mathbf{y}) = \sum_{d=1}^{D} k_{q_d}(\mathbf{x}_d, \mathbf{y}_d) = \sum_{d=1}^{D} \varphi_{q_d}(\mathbf{x}_d)^\top \mathbf{K}_d \varphi_{q_d}(\mathbf{y}_d) = \varphi_q(\mathbf{x})^\top \mathbf{K} \varphi_q(\mathbf{y}), \qquad (9)$$

where $q_d \in \mathcal{Q}_{N_d}^1$, $\varphi_{q_d}$ is the 1-of-$N_d$ coding of dimension $d$, $\mathbf{K}_d$ is the $N_d \times N_d$ Gram matrix of dimension $d$, $\varphi_q$ is the concatenation of the $D$ mappings $\varphi_{q_d}$, and $\mathbf{K}$ is a $(\sum_d N_d) \times (\sum_d N_d)$ block-diagonal matrix of $\mathbf{K}_1, \ldots, \mathbf{K}_D$. The benefits of AQK are twofold. First, the explicit feature space is reduced from $N = \prod_d N_d$ to $N' = \sum_d N_d$. Second, the number of parameters to learn in $\mathbf{K}$ is now only $\sum_d N_d^2$ instead of $N^2$. The compression ratio is unchanged since $\log_2(N) = \sum_d \log_2(N_d)$. To learn $\mathbf{K}$ in Eq. (9), we simply set the off-block-diagonal elements of $G_{t'}$ to zero in each iteration, and iteratively update $\mathbf{K}$ as describe in Sec. 3.2. To optimize a product quantizer, we iterate the optimization of each 1d quantizer $q_d$ following Sec. 3.3, while fixing $q_c$ for $c \neq d$. This leads to using the following energy $E_j$ for a pair $(\mathbf{x}_j, \mathbf{y}_j)$:

$$E_{j,d}(q_d(\mathbf{x}_{j,d}), q_d(\mathbf{y}_{j,d})) = \max\left(0, \mu_{j,d} - l_j \mathbf{K}_d(q_d(\mathbf{x}_{j,d}), q_d(\mathbf{y}_{j,d}))\right), \qquad (10)$$

where $\mu_{j,d} = 1 - l_j \sum_{c \neq d} \mathbf{K}_c(q_c(\mathbf{x}_c), q_c(\mathbf{y}_c))$ acts as an adaptive margin.

**Block quantized kernels (BQK).** Although the additive assumption in AQK greatly reduces the number of parameters, it is also very restrictive, as it assumes independent data dimensions. A simple way to extend additive quantized kernels to model the inter-dependencies of dimensions is to allow the off-diagonal elements of $\mathbf{K}$ in Eq. (9) to be nonzero. As a trade-off between a block-diagonal (AQK) and a full matrix, in this work we also consider the grouping of the feature dimensions into $B$ blocks, and only learn off-block-diagonal elements within each block, leading to Block Quantized Kernels (BQK). In this way, assuming $\forall d \quad N_d = n$, the number of parameters in $\mathbf{K}$ is $B$ times smaller than for the full matrix. As a matter of fact, many features such as SIFT descriptors exhibit block structure. SIFT is composed of a $4 \times 4$ grid of 8 orientation bins. Components within the same spatial cell correlate more strongly than others and, thus, only modeling those jointly may prove sufficient. The optimization of $\mathbf{K}$ and $q$ are straightforwardly adapted from the AQK case.

**Additional parameter sharing.** Commonly, the different dimensions of a descriptor are generated by the same procedure and hence share similar properties. This results in block matrices $\mathbf{K}_1, \ldots, \mathbf{K}_D$ in AQK that are quite similar as well. We propose to exploit this observation and share the kernel matrix for groups of dimensions, further reducing the number of parameters. Specifically, we cluster dimensions based on their variances into $G$ equally sized groups and use a single block matrix for each group. During optimization, dimensions sharing the same block matrix can conveniently be merged, i.e. $\varphi_q(\mathbf{x}) = [\sum_{d \text{ s.t. } \mathbf{K}_d = \mathbf{K}_1'} \varphi_{q_d}(\mathbf{x}_d), \ldots, \sum_{d \text{ s.t. } \mathbf{K}_d = \mathbf{K}_G'} \varphi_{q_d}(\mathbf{x}_d)]$, and then $\mathbf{K} = \text{diag}(\mathbf{K}_1', \ldots, \mathbf{K}_G')$ is learnt following the procedure already described for AQK. Notably, the quantizers themselves are not shared, so the kernel still adapts uniquely to every dimension of the data, and the optimization of quantizers is not changed either. This parameter sharing strategy can be readily applied to BQK as well.

## 4 Results

We now present our experimental results, starting with a description of our protocol. We then explore parameters and properties of our kernels (optimization of quantizers, explicit feature maps). Finally, we compare to the state-of-the-art in performance and compactness.

**Dataset and evaluation protocol.** We evaluate our method using the dataset of Brown et al. [5]. It contains three sets of patches extracted from Liberty, Notre Dame and Yosemite using the Difference of Gaussians (DoG) interest point detector. The patches are rectified with respect to the scale

| | Initial | Optimized |
|---|---|---|
| Uniform | 24.84 | 21.68 |
| Adaptive | 25.99 | 25.70 |
| Adaptive+ | 14.62 | 14.29 |

Table 1: Impact of quantization optimization for different quantization strategies

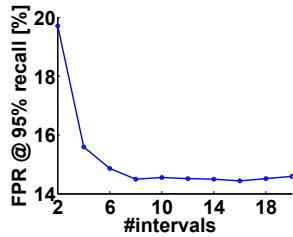

Figure 1: Impact of $N$, the number of quantization intervals

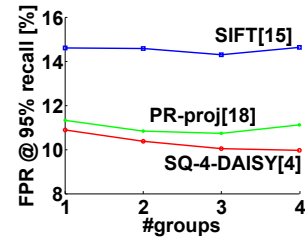

Figure 2: Impact of $G$, the number of dimension groups

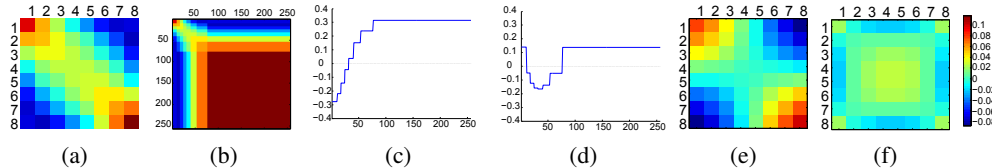

Figure 3: Our learned feature maps and additive quantized kernel of a single dimension. (a) shows the quantized kernel in index space, while (b) is in the original feature space for the first quantizer. (c,d) show the two corresponding feature maps, and (e,f) the related rank-1 kernels.

and dominant orientation, and pairwise correspondences are computed using a multi-view stereo algorithm. In our experiments, we use the standard evaluation protocol [5] and state-of-the-art descriptors: SIFT [15], PR-proj [21] and SQ-4-DAISY [4]. $M$=500k feature pairs are used for training on each dataset, with as many positives as negatives. We report the false positive rate (FPR) at $95\%$ recall on the test set of 100k pairs. A challenge for this dataset is the bias in local patch appearance for each set, so a key factor for performance is the ability to generalize and adapt across sets.

Below, in absence of other mention, AQKs are trained for SIFT on Yosemite and tested on Liberty.

**Interval quantization and optimization.** We first study the influence of initialization and optimization on the generalization ability of the interval quantizers. For initialization, we have used two different schemes: a) *Uniform quantization*, *i.e.* the quantization with equal intervals; b) *Adaptive quantization*, *i.e.* the quantization with intervals with equal number of samples. In both cases, it allows to learn a first kernel matrix, and we can then iterate with boundary optimization (Sec. 3.3). Typically, convergence is very fast (2-3 iterations) and takes less than 5 minutes in total (*i.e.*, about 2s per feature dimension) with 1M nodes. We see in Table 1 that uniform binning outperforms the adaptive one and that further optimization benefits the uniform case more. This may seem paradoxical at first, but this is due to the train/test bias problem: intervals with equal number of samples are very different across sets, so refinements will not transfer well. Hence, following [7], we first normalize the features with respect to their rank, separately for the training and test sets. We refer to this process as *Adaptive+*. As Table 1 shows, not only does it bring a significant improvement, but further optimization of the quantization boundaries is more beneficial than for the Adaptive case. In the following, we thus adopt this strategy.

**Number of quantization intervals.** In Fig. 1, we show the impact of the number of intervals $N$ of the quantizer on the matching accuracy, using a single shared kernel submatrix ($G = 1$). This number balances the flexibility of the model and its compression ratio. As we can see, using too few intervals limits the performance of QK, and using too many eventually leads to overfitting. The best performance for SIFT is obtained with between $8$ and $16$ intervals.

**Explicit feature maps.** Fig. 3a shows the additive quantized kernel learnt for SIFT with $N = 8$ and $G = 1$. Interestingly, the kernel has negative values far from the diagonal and positive values near the diagonal. This is typical of stationary kernels: when both features have similar values, they contribute more to the similarity. However, contrary to stationary kernels, diagonal elements are far from being constant. There is a mode on small values and another one on large ones. The second one is stronger: *i.e.*, the co-occurrence of large values yields greater similarity. This is consistent with the voting nature of SIFT descriptors, where strong feature presences are both rarer and more informative than their absences. The negative values far from the diagonal actually penalize inconsistent observations, thus confirming existing results [12]. Looking at the values in the original space in Fig. 3b, we see that the quantizer has learnt that fine intervals are needed in the lower

| Descriptor | Kernel | Dimensionality | Train on Yosemite | | Train on Notredame | | Mean |
|---|---|---|---|---|---|---|---|
| | | | Notredame | Liberty | Yosemite | Liberty | |
| SIFT[15] | Euclidean | 128 | 24.02 | 31.34 | 27.96 | 31.34 | 28.66 |
| SIFT[15] | $\chi^2$ | 128 | 17.65 | 22.84 | 23.50 | 22.84 | 21.71 |
| SIFT[15] | AQK(8) | 128 | 10.72 | 16.90 | 10.72 | 16.85 | 13.80 |
| SIFT[15] | AQK(8) | 256 | 9.26 | 14.48 | 10.16 | 14.43 | 12.08 |
| SIFT[15] | BQK(8) | 256 | **8.05** | **13.31** | **9.88** | **13.16** | **11.10** |
| SQ-4-DAISY [4] | Euclidean | 1360 | 10.08 | 16.90 | 10.47 | 16.90 | 13.58 |
| SQ-4-DAISY [4] | $\chi^2$ | 1360 | 10.61 | 16.25 | 12.19 | 16.25 | 13.82 |
| SQ-4-DAISY [4] | SQ [4] | 1360 | 8.42 | 15.58 | 9.25 | 15.58 | 12.21 |
| SQ-4-DAISY [4] | AQK(8) | $\leq$1813 | **4.96** | **9.41** | **5.60** | **9.77** | **7.43** |
| PR-proj [21] | Euclidean[21] | <64 | 7.11 | 14.82 | 10.54 | 12.88 | 11.34 |
| PR-proj [21] | AQK(16) | $\leq$102 | **5.41** | **10.90** | **7.65** | **10.54** | **8.63** |

Table 2: Performance of kernels on different datasets with different descriptors. AQK(N) denotes the additive quantized kernel with $N$ quantization intervals. Following [6], we report the False positive rate (%) at 95% recall. The best results for each descriptor are in bold.

values, while larger ones are enough for larger values. This is consistent with previous observations that the contribution of large values in SIFT should not grow proportionally [2, 18, 14].

In this experiment, the learnt kernel has rank 2. We show in Fig. 3c, 3d, 3e and 3f the corresponding feature mappings and their associated rank 1 kernels. The map for the largest eigenvalue (Fig. 3c) is monotonous but starts with negative values. This impacts dot product significantly, and accounts for the above observation that negative similarities occur when inputs disagree. This rank 1 kernel cannot allot enough contribution to similar mid-range values. This is compensated by the second rank (Fig. 3f).

**Number of groups.** Fig. 2 now shows the influence of the number of groups $G$ on performance, for the three different descriptors ($N = 8$ for SIFT and SQ-4-DAISY, $N = 16$ for PR-proj). As for intervals, using more groups adds flexibility to the model, but as less data is available to learn each parameter, over-fitting will hurt performance. We choose $G = 3$ for the rest of the experiments.

**Comparison to the state of the art.** Table 2 reports the matching performance of different kernels using different descriptors, for all sets, as well as the dimensionality of the corresponding explicit feature maps. For all three descriptors and on all sets, our quantized kernels significantly and consistently outperform the best reported result in the literature. Indeed, AQK improves the mean error rate at 95% recall from 28.66% to 12.08% for SIFT, from 13.58% to 7.43% for SQ-4-DAISY and from 11.34% to 8.63% for PR-proj compared to the Euclidean distance, and about as much for the $\chi^2$ kernel. Note that PR-proj already integrates metric learning in its design ([21] thus recommends using the Euclidean distance): as a consequence our experiments show that modelling non-linearities can bring significant improvements. When comparing to *sparse quantization* (SQ) with hamming distance as done in [4], the error is significantly reduced from 12.21% to 7.43%. This is a notable achievement considering that [4] is the previous state of the art.

The SIFT descriptor has a grid block design which makes it particularly suited for the use of BQK. Hence, we also evaluated our BQK variant for that descriptor. With BQK(8), we observed a relative improvement of 8%, from 12.08% for AQK(8) to 11.1%.

We provide in Fig. 4 the ROC curves for the three descriptors when training on Yosemite and testing on Notre Dame and Liberty. These figures show that the improvement in recall is consistent over the full range of false positive rates. For further comparisons, our data and code are available online.[3]

**Compactness of our kernels.** In many applications of feature matching, the compactness of the descriptor is important. In Table 3, we compare to other methods by grouping them according to their memory footprint. As a reference, the best method reported in Table 2 (AQK(8) on SQ-4-DAISY) uses 4080 bits per descriptor. As expected, error rates increase as fewer bits are used, the original features being significantly altered. Notably, QKs consistently yield the best performance in all groups. Even with a crude binary quantization of SQ-4-DAISY, our quantized kernel outperform the state-of-the-art SQ of [4] by 3 to 4%. When considering the most compact encodings ($\leq$ 64 bits), our AQK(2) does not improve over BinBoost [22], a descriptor designed for extreme compactness, or the product quantization (PQ [13]) encoding as used in [21]. This is because our current framework does not yet allow for joint compression of multiple dimensions. Hence, it is unable to use less

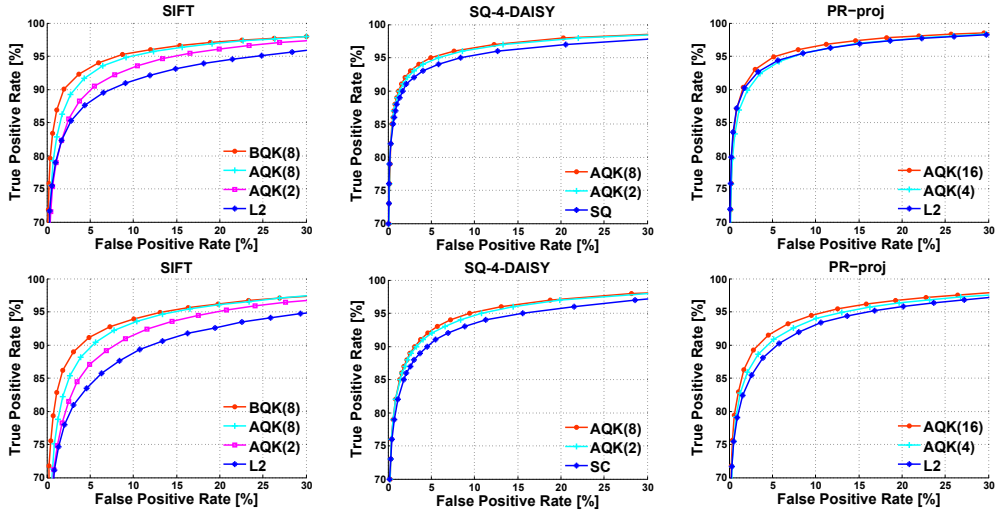
Figure 4: ROC curves when evaluating Notre Dame (top) and Liberty (bottom) from Yosemite

| Descriptor | Encoding | Memory (bits) | Train on Yosemite | | Train on Notredame | | Mean |
|---|---|---|---|---|---|---|---|
| | | | Notredame | Liberty | Yosemite | Liberty | |
| SQ-4-DAISY [4] | SQ [4] | 1360 | 8.42 | 15.58 | 9.25 | 15.58 | 12.21 |
| SQ-4-DAISY [4] | AQK(2) | 1360 | **5.86** | **10.81** | **6.36** | **10.94** | **8.49** |
| SIFT[15] | AQK(8) | 384 | 9.26 | 14.48 | 10.16 | 14.43 | 12.08 |
| PR-proj [21] | Bin [21] | 1024 | 7.09 | 15.15 | 8.5 | 12.16 | 10.73 |
| PR-proj [21] | AQK(16) | <256 | **5.41** | **10.90** | **7.65** | **10.54** | **8.63** |
| SIFT[15] | AQK(2) | 128 | 14.62 | 19.72 | 15.65 | 19.45 | 17.36 |
| PR-proj [21] | Bin [21] | 128 | 10.00 | 18.64 | 13.41 | 16.39 | 14.61 |
| PR-proj [21] | AQK(4) | <128 | **7.18** | **13.02** | **10.29** | **13.18** | **10.92** |
| BinBoost[22] | BinBoost[22] | 64 | 14.54 | 21.67 | 18.97 | 20.49 | 18.92 |
| PR-proj [21] | AQK(2) | <64 | 14.80 | 20.59 | 19.38 | 22.24 | 19.26 |
| PR-proj [21] | PQ [21] | 64 | 12.91 | 20.15 | 19.32 | 17.97 | 17.59 |
| PR-proj [21] | PCA+AQK(4) | 64 | **10.74** | **17.46** | **14.44** | **17.60** | **15.06** |

Table 3: Performance comparison of different compact feature encoding. The number in the table is reported as False positive rate (%) at 95% recall. The best results for each group are in bold.

than 1 bit per original dimension, and is not optimal in that case. To better understand the potential benefits of decorrelating features and joint compression in future work, we pre-processed the data with PCA, projecting to 32 dimensions and then using AQK(4). This simple procedure obtained state-of-the-art performance with 15% error rate, now outperforming [22] and [21].

Although QKs yield very compact descriptors and achieve the best performance across many experimental setups, the computation of similarity values is slower than for competitors: in the binary case, we double the complexity of hamming distance for the $2 \times 2$ table look-up.

## 5 Conclusion

In this paper, we have introduced the simple yet powerful family of quantized kernels (QK), and presented an efficient algorithm to learn its parameters, *i.e.* the kernel matrix and the quantization boundaries. Despite their apparent simplicity, QKs have numerous advantages: they are very flexible, can model non-linearities in the data and provide explicit low-dimensional feature mappings that grant access to the Euclidean geometry. Above all, they achieve state-of-the-art performance on the main visual feature matching benchmark. We think that QKs have a lot of potential for further improvements. In future work, we want to explore new learning algorithms to obtain higher compression ratios – *e.g.* by jointly compressing feature dimensions – and find the weight sharing patterns that would further improve the matching performance automatically.

## Acknowledgements

We gratefully thank the KIC-Climate project Modeling City Systems.

## Footnotes

[1]Metric learning can be kernelized, but then one has to choose the kernel.

[2]As any continuous function on an interval is the uniform limit of a series of piecewise constant functions, this assumption does not inherently limit the flexibility of the family.

[3]See: http://www.vision.ee.ethz.ch/~qind/QuantizedKernel.html

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
