[Reviews · NeurIPS 2014]

Submitted by Assigned_Reviewer_8

Summary:The paper attempts to learn quantized kernel functions. It accomplishes that by discretizing real valued features and then learn
kernel function.
They propose a formulation which simultaneously tries to optimise the kernel function and the
discretization function. The optimisation over kernel function reduces that of finding a Kernel matrix.
The joint optimisation problem of finding the optimal kernel matrix and
discretization function is intractable, and is solved by alternate minimisation. Experimental results are provided.

The motivation for the problem is not clear. More specifically it is not clear what extra benefits does the discretising function brings. Indeed in lines 54-57 the paper argues the benefits
of learning discrete kernels.
What i understand from that argument is that quantisation can be beneficial if one wants a small memory
footprint of the Kernel matrix. If this is correct then recent work on Binary embeddings(see e.g. [1] below) where one tries to
quantise high dimensional feature vectors by a smaller vector consisting of \{1,-1\} entries,
is relevant. Clearly then a comparison
is warranted between the proposed approached and Binary embeddings.

One way to discretize the kernels is to replace the entries of Kernel matrices by 1,-1, or 0. This was attempted before(see [2] below). It would be nice if the paper can discuss this work as part of related work.

Given the understanding in the field of Kernel learning, formulation (5) looks straightforward. Moreover it is
not tractable and the paper proposes alternate minimisation which without convergence analysis appear as a heuristic. It would have been nice if the algorithm had some guarantees.
This makes the contribution to machine learning, the key focus of NIPS, appear very modest.

Minor point:
The function $q(x)$ is not defined properly. If i understand correctly. It the author(s) mean
" q(x)
is function assigning x a value in \{1,\ldots,N\}$.
Often [a,b] means a closed interval.

[1] Gong, Yunchao, Kumar, Sanjiv, Verma, Vishal, and Lazebnik,
Svetlana. Angular quantization-based binary codes
for fast similarity search. In Advances in Neural Information
Processing Systems, 2012.

[2]Dimitris Achlioptas, Frank McSherry, Bernhard Schölkopf: Sampling Techniques for Kernel Methods. 335-342
(see section 3).

After reading the author feedback
--------------------
Concerns about prior work and benchmarking are addressed to some extent. It would have been helpful if the authors could
have shown results on standard learning task such as classification and benchmarked them against [2].

Summary: The contribution to machine learning looks limited. Key references seem to have gone un-noticed.
I would be happy to reconsider my review if the authors clearly address the issues raised in the review. More specifically the issues related to motivation and prior work needs to be convincingly addressed.

Submitted by Assigned_Reviewer_17

This paper proposes to learn quantized kernel for feature matching in computer version tasks. It first maps each original feature into an index space and then additive and blocked quantized kernel is learned in the index space. The authors show empirically the proposed method is favorable comparing to other state-of-the-art.

Technically, this is an interesting paper. The optimization method of interval quantization optimization potentially could be used into other learning problems.

Most of my concerns about the proposed method, e.g. how to initialize, the sensitivity of N, etc, have been discussed in the section of experimental results.
This is a well-written paper. Most of the technical description is clear, except the section of interval quantization optimization. What is the meaning of E_j(q(x_i),q(x_j)) in line 189? And also [0, E_j(2,2)] in line 199?

Finally, I am not from the computer version community and do not have much experience with CV datasets. But, I guess this proposed method most probably might not improve classification accuracy on a lot of UCI benchmark datasets. Can the authors share some insights on why does quantization help a lot here?

Minors:
In experiment, the final dimensionality D of learning instance should be reported.

In case the authors are not aware, the MKL approach can learn with thousands of kernels.
Ultra-Fast Optimization Algorithm for Sparse Multi Kernel Learning, ICML 2011.
Summary: Technically, this is an interesting paper. The experimental evaluation is convincing.

Submitted by Assigned_Reviewer_35

The paper provides a new algorithm for learning quantized kernels: kernels based on piecewise constant mappings of the features into a finite index space. The main motivation and application is identifying similar pairs in vision tasks. The method achieves significant improvement over state-of-the-art on a task of identifying patch pairs between images.

Pros:
1. The experimental results, while confined to a specific task, are very strong. The method, while slightly slower, improves ginificantly over state-of-the-art in several metrics and experimental conditions (e.g. explicit feature maps, compact codes).
2. The experimental analysis of the learned kernels is deep and insightful to the nature of the specific problem and the way quantized kernels help approach it.
3. The approach is novel. Quantized and compact descriptors have been hugely successful in vision applications. This paper takes these ideas into the realm of kernels and offers an interesting way to adaptively learn a quantized kernel.

Cons:
1. The paper is hard to follow, especially the technical details in section 3. Some examples:
a. The definition of a quantizer is cumbersome, instead of just a function q:R^D -> {1,...,N}.
b. The optimization algorithm is presented for the one-dimensional case with 2 labels. That is in itself a very easy problem to solve, with no need for the more complicated machinery as brought in the paper, which made it harder to follow for me. Later, there is mention of how to (very approximately) reduce the multi-dimensional, multi-label case to this simple case. This whole line of thought should be made more explicit and easier to follow.
c. Last line of page 4: isn't b_1 the same for all t? If so, which t to choose?
d. Should I conclude from lines 246-247 that the AQK kernel is actually a diagonal kernel?
e. The logic of the subsection about sharing kernel matrices is particularly hard to follow, at least for someone not very familiar with these methods.
I think the paper could benefit from an "algorithm" box stating explicitly how to solve the full dimensional, arbitrary N case, for both AQK and BQK.

Misc.:
1. The problem attempts to classify pairs in negative and positive pairs. Obviously not all pairs can be used for large datasets, especially since most will be negative. How are the pairs chosen in the experiments?
2. In other similarity learning tasks it is common to learn a bilinear form such that a triplet order is preserved instead of positive and negative pairs. For example in "Learning a Distance Metric from a Network" by Shaw et al., NIPS 2011. Has this been tried in the context of the tasks used in this paper?
3. Line 315 and table 1: it seems that for Adaptive+ the Optimized version does NOT improve performance significantly. It is on the same order as in the Adaptive case, where the authors claim there is no improvement.
Summary: The paper presents a novel and potentially very useful idea, with strong experimental results on a specific computer vision patch matching task. The paper isn't written very clearly, and is dense with details and various ideas, making it harder to follow at times.

Submitted by Assigned_Reviewer_41

This paper proposes a new kind of kernel learning algorithm, which lies between non-parametric kernel learning and multiple kernel learning. Basically the gram matrix is directly learnt, with the only restriction that the kernel is piecewise constant. This particular formulation defines a rather flexible family of kernels, and yet the optimization problem can be carried out rather effectively. The paper is well written and easy to follow. The technical part seems to be sound, but I didn’t check it thoroughly. The empirical study is convincing.
Summary: This paper proposes a new kind of kernel learning algorithm, which lies between non-parametric kernel learning and multiple kernel learning. Basically the gram matrix is directly learnt, with the only restriction that the kernel is piecewise constant.
Author Feedback
Author rebuttal: We would like thank the reviewers for their reviews. Below we address the main questions.

Difference to missing references [1,2] (R8) and explicit kernel features (MR6)

Those works are significantly different from ours, but indeed share some properties with our approach, which we will discuss in the paper. [1,2] and, eg, Vedaldi and Zisserman [21], all aim at approximating a known and fixed kernel to speed up kernel evaluation and/or kernel methods.
- In [1], the authors obtain a compact binary representation of the data to approximate the cosine/Euclidean kernel.
- In [2], the authors approximate the Gram matrix of training data by limiting its entries to {-1,0,1}.
- Explicit kernel features (such as [21]) provide low-dimensional approximations of various standard kernels.
As good as those methods are, they cannot perform better than the pre-defined kernel they try to approximate. Instead, in our work, we do not assume a fixed kernel but want to find the best kernel for our matching task. We propose a novel family of kernels (see also motivations below) which is parametrized by (a) a quantization function of the input space to N elements and (b) the Gram matrix comparing those N elements. Our work is about learning (a) and (b) for a joint objective (eq 5). As benefits of our model, (a) leads to explicit compact descriptors and efficient kernel evaluation; and (b) indeed limits the number of unique values of any Gram matrix to at most N^2, hence is reminiscent of [2].

Unclear motivations for discretization (R8)

The main motivation is to parametrize a very large and flexible family of kernels in a convenient way for learning. As a matter of fact, although our family of quantized kernels is dense in the space of smooth kernels (l.107), we show how learning can still be performed. The small memory footprint and efficient kernel evaluation are by-product benefits.

R8: Equation (5) is standard; heuristic method proposed

Eq (5) is indeed a classical objective based on hinge loss, except we also optimize it w.r.t the quantizer. This is the main technical contribution of the paper (sec. 3.3 and 3.4). As R8 recognizes, eq (5) is indeed intractable, so approximate optimization is inevitable.

R17: Why quantization helps here? Application to classification?

Quantization alone does not help for performance. The key is to learn the kernel. We believe that our quantized kernels could be potentially useful for other applications, because using good similarity measures has a great impact on many systems.

R17: MKL can combine thousands of kernels

Indeed, quantized kernels can be seen as linear combinations of indicator kernels (l.110). However, since every point is potentially in its own quantization cell, the number of kernels implicitly combined with our method is quadratic in the number of data points, ie billions of kernels. This number is too large for generic MKL solvers. In essence, specialized techniques like ours exploit the specific nature of the base kernels to make it feasible.

R35: Unclear algorithm

We will simplify the description of our algorithm. In 1d and binary case, b_1 is the quantization boundary that we look for. Our algorithm finds the consecutive values v_t and v_{t+1} where the best quantization index change should happen, hence we then use the mean as boundary. Equivalently, we can evaluate the energy at each of these means by exploiting efficient partial updates. This also has linear time complexity.

R35: AQK diagonal (l.246-247)? Sharing kernels?

AQK is not diagonal, it is an ND-by-ND block-diagonal matrix where each block is a full N-by-N kernel matrix for each of the D dimensions. Sharing kernels means that those N-by-N blocks can be shared across dimensions, ie repeated in the larger matrix. But the individual quantizers need not be shared, as long as they have the same number of quantization bins. We will clarify those elements.

R35: Misc. remarks

1. The pairs are part of the dataset released by Brown et al.
2. Using triplets is indeed common in similarity matching. In our case, they would yield ternary potentials in the energy function, which are even harder to optimize than pairwise ones.
3. We will rephrase this sentence.

Finally, we will take into account all the other comments: more standard definition of discrete sets (R8,R35), clearer notations (R17), etc. We thank again the reviewers for finding our work well-written (R17,R41), novel (R35,R41), interesting (R17,R35) and obtaining convincing state-of-the-art results (R17,R35,R41).